# Redox-freezing and nucleation of diamond via magnetite formation in the Earth's mantle

Dorrit E. Jacob[1], Sandra Piazolo[1], Anja Schreiber[2] & Patrick Trimby[3]

Diamonds and their inclusions are unique probes into the deep Earth, tracking the deep carbon cycle to >800 km. Understanding the mechanisms of carbon mobilization and freezing is a prerequisite for quantifying the fluxes of carbon in the deep Earth. Here we show direct evidence for the formation of diamond by redox reactions involving FeNi sulfides. Transmission Kikuchi Diffraction identifies an arrested redox reaction from pyrrhotite to magnetite included in diamond. The magnetite corona shows coherent epitaxy with relict pyrrhotite and diamond, indicating that diamond nucleated on magnetite. Furthermore, structures inherited from h-$Fe_3O_4$ define a phase transformation at depths of 320–330 km, the base of the Kaapvaal lithosphere. The oxidation of pyrrhotite to magnetite is an important trigger of diamond precipitation in the upper mantle, explaining the presence of these phases in diamonds.

[1] Australian Research Council Centre of Excellence for Core to Crust Fluid Systems and Department of Earth and Planetary Sciences, Macquarie University, North Ryde, New South Wales 2109, Australia. [2] GeoForschungsZentrum Potsdam, Telegrafenberg C155, Potsdam D-14473, Germany. [3] Australian Centre for Microscopy and Microanalysis, The University of Sydney, Sydney, New South Wales 2006, Australia. Correspondence and requests for materials should be addressed to D.E.J. (email: dorrit.jacob@mq.edu.au).

The Earth's mantle contains 200 µg g$^{-1}$ sulfur on average[1]. Despite this element's rarity, iron-nickel (FeNi) sulfides form by far the most abundant inclusions in natural diamond[2,3]. This extreme overabundance in comparison to the mantle xenolith record is enigmatic, particularly because the mobility of low-volume sulfide melts in the mantle is limited[4].

An increasing body of evidence indicates that the majority of diamonds form from mobile carbon-bearing fluids and melts in the Earth's mantle[5]. Clues for diamond formation mechanisms are found in minerals and fluids included in various diamond species, which provide different perspectives of the same process. Fast-growing fibrous diamonds directly sample mantle metasomatic fluids in abundant micro-inclusions[6]. In contrast, rapidly formed polycrystalline diamond aggregates encase a range of solid products representative of their formation process[7]. Combined evidence from these rapidly formed diamond species emphasizes the importance of redox gradients as a controlling parameter for the immobilization of carbon as diamond in the mantle[6,8,9].

The most effective redox couple in the Earth's mantle consists of carbon and iron, which controls the onset of carbonate melting with depth by reduction of $Fe^{3+}$ and oxidation of graphite or diamond[9]. Conversely, oxidation of $Fe^{2+}$ may lead to freezing of mobile carbon species in the form of graphite or diamond. $Fe^{2+}$ in FeNi sulfides in the Earth's mantle thus present a reservoir with considerable oxidation potential. While rare in the mantle overall[1], billions of years of subduction[10,11] have created patches that are highly enriched in crustal material that contain abundant sulfides, carbon and volatiles and provide a very reactive environment in the Earth's mantle.

In this work we investigate the microstructure and composition of FeNi-sulfide inclusions in a polycrystalline diamond aggregate that display a nanogranular magnetite reaction corona. This assemblage demonstrates that diamond formed and nucleated by a redox reaction involving the diamond-forming fluid and the FeNi sulfide that formed magnetite and diamond. Epitaxy between sulfide, the magnetite corona and diamond host shows that diamond nucleated on the magnetite, establishing syngenicity between the phases, and demonstrating magnetite formation at the expense of the FeNi sulfide to diamond formation. Within the magnetite corona, characteristic and systematic orientation relationships between the magnetite grains are indicative of a phase transformation from a high-pressure h-$Fe_3O_4$ phase, which is stable only above 10 GPa, placing the origin of this assemblage unequivocally at the base of the subcratonic lithosphere. This study provides direct evidence for the involvement of FeNi sulfides in diamond formation and emphasizes the importance of locally sulfide-enriched areas in the Earth's mantle as diamond factories.

## Results

**Petrography of the specimen.** The Orapa diamond mine in Botswana is characterized by a particularly dominant eclogitic subduction component in its diamond inclusion suite[12]. An electron-transparent foil of ca. 150 nm thickness was cut from a polycrystalline diamond aggregate containing eclogite suite inclusions[8] from this locality. The foil consists of a single undeformed diamond crystal encasing two angular pyrrhotite inclusions both partially rimmed by a nanocrystalline corona of magnetite ($Fe_3O_4$) and connected by a thin pyrrhotite veinlet (Fig. 1). Microstructure and crystallography of the minerals in this foil were studied using Transmission Kikuchi Diffraction (TKD)[13], a novel technique for diffraction analysis and automated orientation mapping in the Scanning Electron

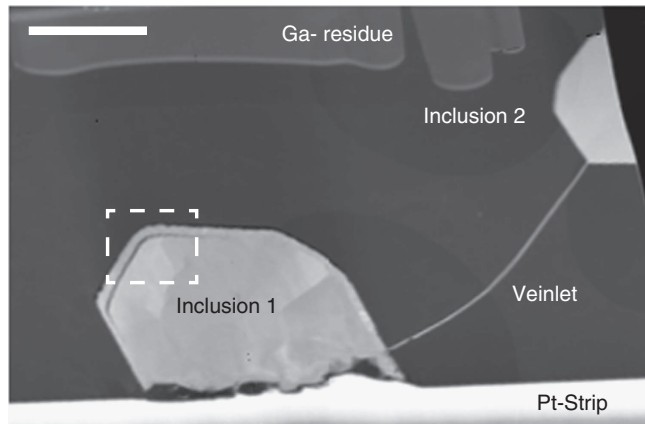

**Figure 1 | Overview of the pyrrhotites included in diamond.** TEM high-angle annular dark field (HAADF) image of the diamond foil showing two pyrrhotite inclusions connected by a pyrrhotite veinlet. The foil is covered by a protective platinum strip and displays some gallium residue from FIB milling. Scale bar, 2 µm.

Microscope (SEM) that achieves ca. 5 nm spatial resolution. Introduced from material sciences, this is the first application to a natural sample.

**Sulfide microstructures.** While TKD is not a very accurate technique for the determination of d-spacings, the Kikuchi patterns of both pyrrhotite grains match closest with a hexagonal high-temperature $Fe_9S_{10}$ structure with $a = 0.343$ and $c = 0.579$ nm. Crystal-plastic deformation is indicated for each by systematic crystallographic orientation changes of up to 5° over 1 µm with gradual lattice bending and distinct subgrain boundaries (Fig. 2b,c) with characteristic orientation dispersion patterns[14] (Fig. 3a,b). In contrast, the host diamond grain is completely undeformed apart from a minor brittle deformation event indicated by the pyrrhotite veinlet (Fig. 1). Brittle deformation occurs at shallower depths than those of plastic deformation; thus this event occurred at a very late stage, possibly upon eruption of the Orapa kimberlite, and is not related to diamond formation.

Plastic deformation microstructures in sulfides very similar to the ones observed here are produced at the onset of dynamic recrystallization at temperatures of ~50% below the melting point of the Fe-S system[15] (>500–550 °C and 0.3 GPa (ref. 16). No sulfide deformation experiments are available under upper mantle conditions, but taking into account the pressure effect on the melting curve of pyrrhotite (ca. +6.2 °C per 0.1 GPa; (ref. 17) extrapolation of the temperatures at surface conditions estimated above indicate ca. 800 °C for pressures of the diamond stability field. These temperatures are well below those required for plastic deformation of diamond (≥1,200 °C and 4 GPa (ref. 18). Therefore, the plastic deformation features shown by the sulfides and their absence in the diamond host indicate that the sulfides were deformed *before* being included into the diamond. Post inclusion annealing within the diamond is unlikely, as this would have affected the diamond and its inclusions equally. This demonstrates that the sulfides are protogenetic and experienced ductile deformation before encapsulation by the diamond.

**Late-stage chalcopyrite exsolutions.** Discrete chalcopyrite grains ($CuFeS_2$, Fig. 2) located along the sulfide subgrain boundaries and Cu-enrichment observed along the outer rims of the pyrrhotite

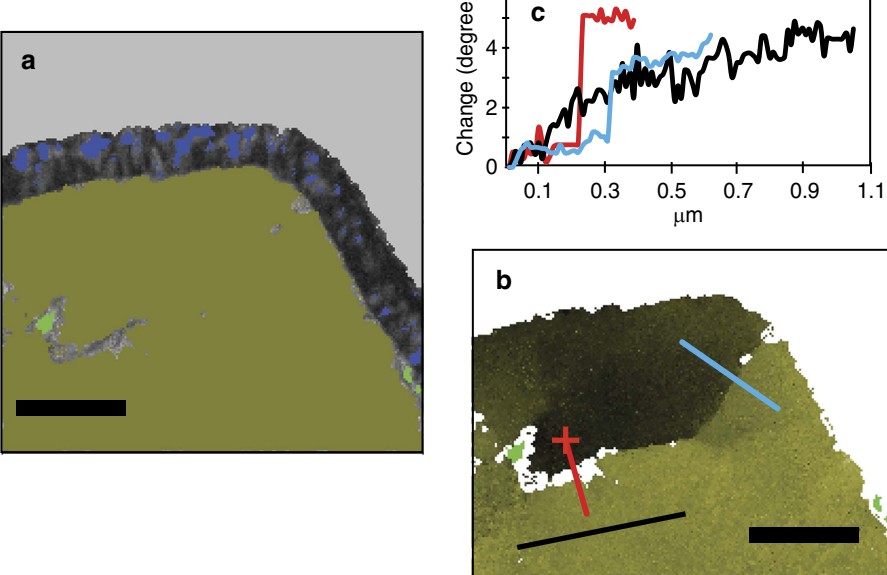

**Figure 2 | Pyrrhotite analysis. (a)** Forescatter electron image overlain by a phase map highlighting the polycrystalline magnetite corona (blue) and two chalcopyrite exsolutions (light green) within pyrrhotite. (**b**) Pyrrhotite inclusion without the magnetite rim (see light green chalcopyrite for orientation), highlighting the relative change in crystallographic orientation within the pyrrhotite by a change of green hues across the inclusion. The lines (red and blue across a subgrain boundary, black within grain) indicate profiles for the quantified orientation change of up to 5° as plotted in (**c**). Scale bar, 0.5 μm.

inclusions (Fig. 3c,d) are late-stage, low-temperature features. Chalcopyrite commonly exsolves from FeNi sulfides upon cooling[19] typically along grain boundaries. Their distribution along subgrain boundaries and the inclusion rim of the sulfides shows that both plastic deformation and corona formation in the sulfide pre-dated Cu exsolution.

**Magnetite corona and evidence for epitaxy.** Coronae of nano-crystalline magnetite ($Fe_3O_4$) about 200 nm wide (Figs 1 and 2) surround both pyrrhotites, but magnetite is spatially very restricted along the rim of inclusion 2 (Fig. 3e,f). Polycrystalline coronae are typical products of incomplete reactions around relict grains[20] and are characterized by epitaxy, namely, a crystallographic orientation aligned to that of the overgrown phase. The larger corona around inclusion 1 shows epitaxy between all three phases, pyrrhotite, magnetite and diamond, with the crystallographic <111> axes of magnetite aligned with the c-axes of both diamond and pyrrhotite (Fig. 4). The coherent epitaxy between relict sulfide, the magnetite reaction rim and host diamond indicates that the magnetite overgrowth formed at the expense of sulfide, after which diamond nucleated on the surface of magnetite, eventually encasing the assemblage and terminating the redox reaction. In contrast, the pyrrhotite of inclusion 2 shows very little magnetite corona formation and the sulfide does not display an epitactic relationship with the diamond host. Thus, in contrast to inclusion 1, inclusion 2 was accidentally entrapped; it was overgrown by the diamond host, but without providing a nucleation substrate.

**Microstructure of the magnetite corona.** The foil was carefully re-thinned by focussed ion beam milling to facilitate more detailed analysis of the magnetite (Fig. 5a). The thinned magnetite corona revealed a distinctive high abundance of twin boundaries arranged in domains of crystallographic orientations (Fig. 5b). Each domain consists of pairs of twins

characterized by a 60° rotation around one of the <111> axes of magnetite. Pole figures depicting the complete crystallographic orientation data for the rim (Fig. 6a,b) show that the distribution is not random, but that several orientation spaces are left unoccupied. These are rimmed by orientations delineating a pattern with the appearance of small 'rings' and 'fences'. These features are characteristic for phase transformations indicating a change in crystal structure between parent and daughter phases. They have been described for phase transitions in metals[21] and in ice[22] and result from a systematic reorientation relationship of parallel directions and planes in both phases. In $Fe_3O_4$ an unquenchable phase transformation from magnetite to h-$Fe_3O_4$ occurs at 10 GPa (ref. 23) with a near-isobaric phase boundary. In high-pressure experiments, nanometre-scale twin lamellae in the retrograde phase along the {311} planes in magnetite are a characteristic record of this transformation, which causes rotation of the <110> direction by 62° with respect to the host crystal lattice[23]. While TKD analysis could not resolve the crystallographic planes of nano-twins in the magnetites, it provides confirmation in 3D that three of the four <110> axes of magnetite coincide (Fig. 6a). This results from a 60° rotation around the <111> axis, very similar to the angle reported by ref. 23. This means that the twinned magnetite grains in the corona are retrograde products of the phase transformation from h-$Fe_3O_4$ to magnetite. The magnetite-h-$Fe_3O_4$ phase boundary at 10 GPa is thus a minimum pressure estimate for this sample, which must therefore originate at ~320–330 km, close to the base of the Kaapvaal subcratonic lithosphere [24]. This is the first time that such a deep minimum pressure of origin for polycrystalline diamond aggregates has been estimated.

## Discussion
Our knowledge about ages and formation of diamond comes primarily from the minerals it includes and relies on the

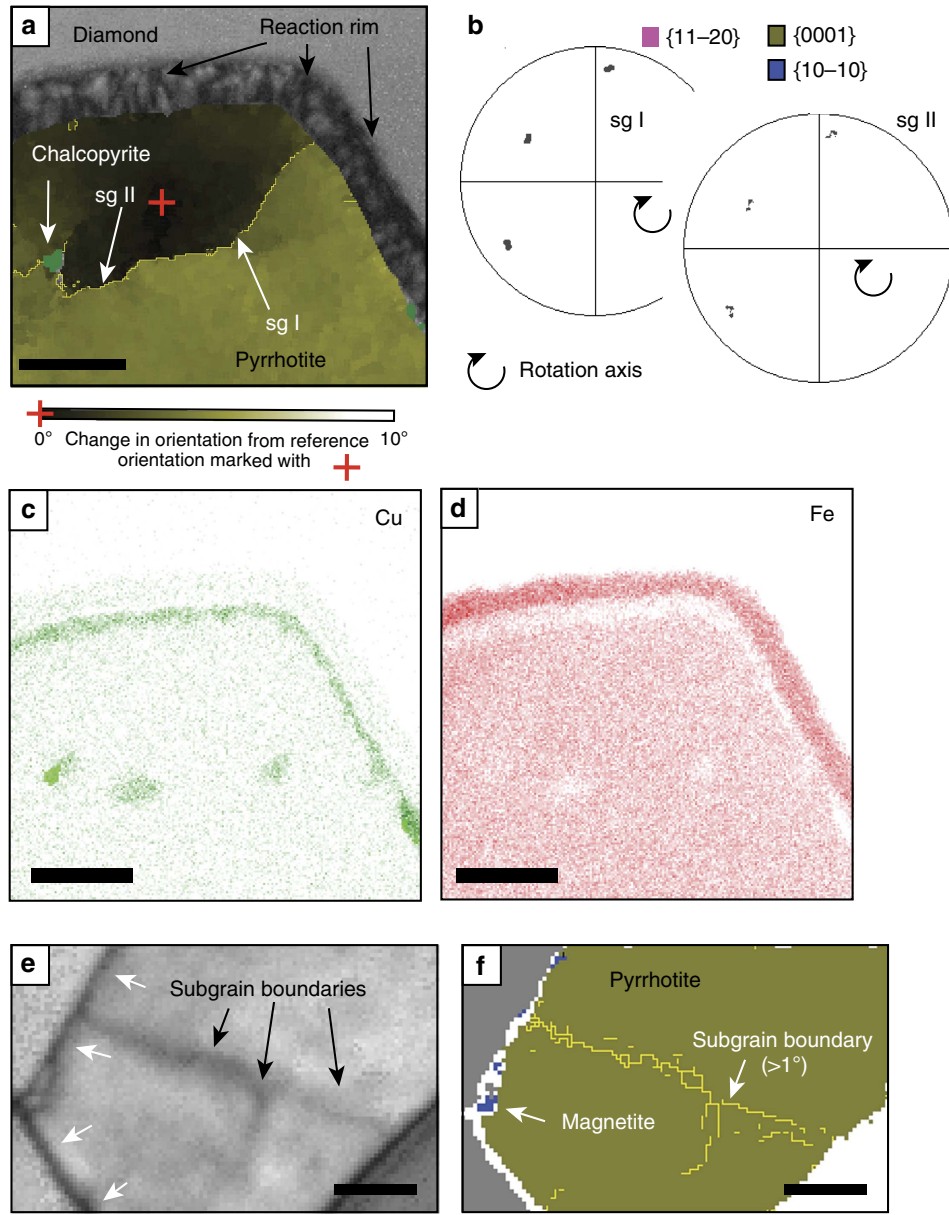

**Figure 3 | Detailed microstructure of the inclusions.** (**a**) Microstructure of inclusion 1 with subgrain boundaries (sg I, sg II). (**b**) Pole figures of dispersion patterns of subgrain boundaries in sulfide are indicative of plastic deformation. The 'boundary trace analysis' technique[33] was used to determine the activated slip system, which is consistent with dispersion patterns across subgrain boundaries. (**c,d**) Element distribution maps for Cu and Fe across the pyrrhotite show distinctive Cu enrichment, and correlated Fe depletion, where chalcopyrite inclusions were identified and an area of Cu enrichment along the rim of the pyrrhotite. (**e,f**) The magnetite on the outside of inclusion 2 is only very small. Scale bar: (**a,c,d**) 0.5 μm; (**e,f**) 0.2 μm.

prerequisite that these minerals are syngenetic. Epitaxy between inclusions and diamond hosts is an unambiguous indicator for syngenicity, but as of yet studies have yielded ambiguous results for inclusion suites within the same diamonds. For example, while olivine inclusions in diamond are generally randomly oriented with respect to the diamond, some olivines within one diamond host can be iso-oriented, implying that these may be fragments of a protogenetic larger crystal[25]. The microstructural observations in this study show that epitactic relationships do indeed identify nucleation centres for diamond and thus establish syngenicity *sensu stricto*. However, as illustrated by the very similar sulfide-magnetite assemblage of inclusion 2 separated only by a few microns from inclusion 1 (Fig. 1), the lack of epitaxy does not discount syngenicity. It simply shows that these phases

did not serve as nucleation centres, but were accidentally encapsuled.

Although sulfides are the most common inclusions in diamond and it has long since been speculated that they may be directly involved in diamond formation[26–28], direct evidence for this has been lacking. Our key sample preserves the critical mineral assemblage and microtexture, thus establishing unambiguously the link between diamond, sulfide and Fe-oxides as partners in and products of an arrested redox reaction. Sulfides are locally enriched in the mantle, where subduction processes govern the flux of sulfur and carbon into the Earth's interior[29]. Subducted compositionally heterogeneous crustal material has been shown to be common over large depth ranges into the lower mantle in modern subduction zones[30,31]. Fossil-subducted crust in the

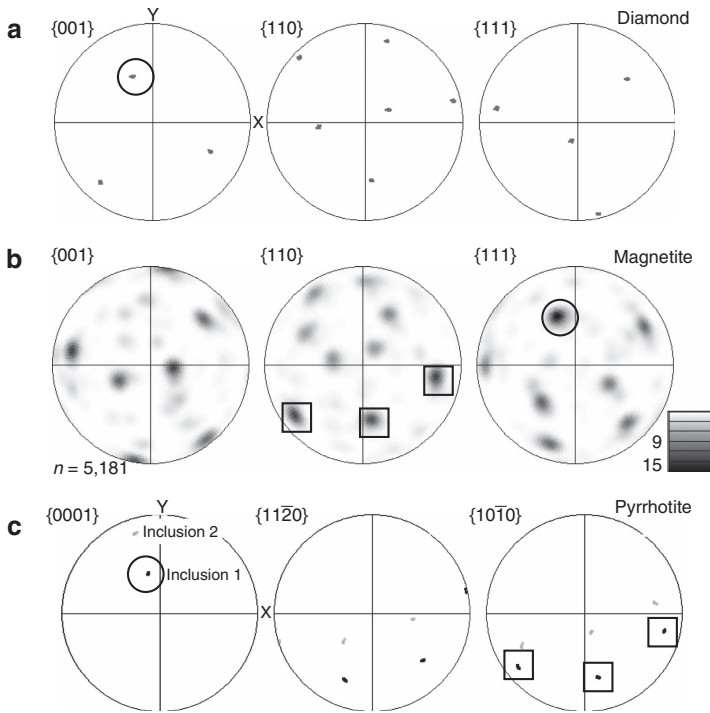

**Figure 4 | Pole figures showing orientation relationships.** Circles depict the epitaxy between one <001> axis of diamond (**a**), the <0001> axis of pyrrhotite in inclusion 1 (**c**) and one of the magnetite <111> axes (**b**). Squares denote the epitaxy between magnetite and pyrrhotite in inclusion 1. One of the {111} axes of magnetite (5181 datapoints) correlates with the {001} axis of inclusion 1 and the crystallographic {001} axis of the host diamond. Pyrrhotite in inclusion 2 (**c**) shows no epitaxy with the host diamond (**a**).

subcratonic lithosphere leaves its legacy in the kimberlite xenolith record[5] and may be particularly abundant at the base of cratonic lithosphere, because the shape of the cratonic roots in the overriding plate plays an active role in causing permanent underplating by subducted ocean crust[32]. These highly reactive areas contain the ingredients and the strong oxidation gradients in close spatial vicinity and have served as 'diamond factories' throughout the major part of Earth's history[10].

## Methods

**Specimen.** The TEM foil is ca. 15 by 10 μm and 0.150–0.200 μm thick, and was cut from a polycrystalline diamond aggregate from the Orapa Mine (Botswana). It contains two angular sulfide inclusions connected by a ca. 0.07-μm-wide sulfide veinlet. The foil was prepared by focused ion beam milling (FIB) in an FEI FIB200 instrument following the methods outlined in ref. 34. After milling, the foil was cut free, extracted and placed flat on a carbon-coated Cu grid without further carbon coating. The sulfides were identified as pyrrhotite by their element composition and their lattice parameters upon selected area electron diffraction in the TEM[8], and TKD analysis revealed best match with the crystal structure of high-temperature hexagonal pyrrhotite ($Fe_9S_{10}$) with $a = 0.343$ and $c = 0.579$ nm.

**Transmission Kikuchi Analysis.** Transmission Kikuchi Analysis (TKD) analysis was performed using a Zeiss Ultra Plus FEG SEM, equipped with an Oxford Instruments Channel 5 EBSD system and a Nordlys-S EBSD detector at the Australian Centre for Microscopy and Microanalysis, The University of Sydney, operated at 1–10 nA and 30 kV at high vacuum. Phase composition was determined using an Oxford Instruments AZtec EDS system with an X-Max 20 mm² silicon drift detector. The TEM foil was mounted using custom-made clamps attached to a standard 70° tilted EBSD sample holder. The SEM stage was tilted toward the EBSD detector by 20° at a working distance of typically 5 mm from the pole piece[13]. After positioning the SEM stage the EBSD detector including phosphorous screen and forescatter detectors was fully inserted.

**Data collection and reduction.** Orientation and element distribution maps were collected simultaneously by stepping the electron beam across the surface in a rectangular grid with a step size of 25 nm. Typically 50–100 X-rays were measured from each point. Diffraction patterns were collected with a resolution of 336 × 256

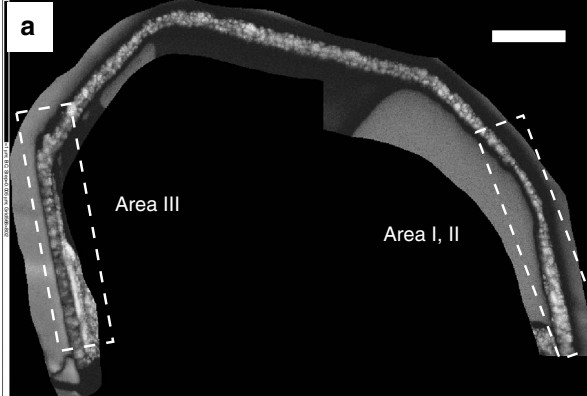

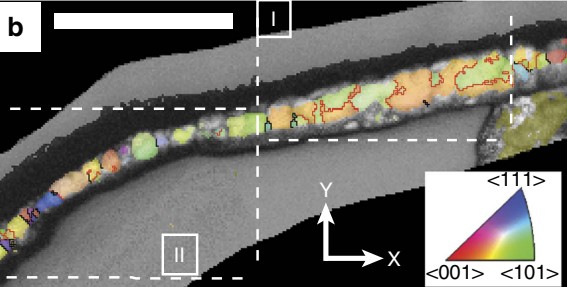

**Figure 5 | Crystallographic analysis of the magnetite corona.** (**a**) Forescatter electron image of the re-thinned magnetite corona. Pyrrhotite was lost during re-thinning. The crystallography of Areas I, II and III was further analysed. (**b**) Crystal preferred orientation of magnetite in Areas I and II, colour-coded according to the legend and the reference frame (bottom right). Grain boundaries are in black, red lines are twin boundaries. (**a**,**b**) Scale bar, 0.5 μm.

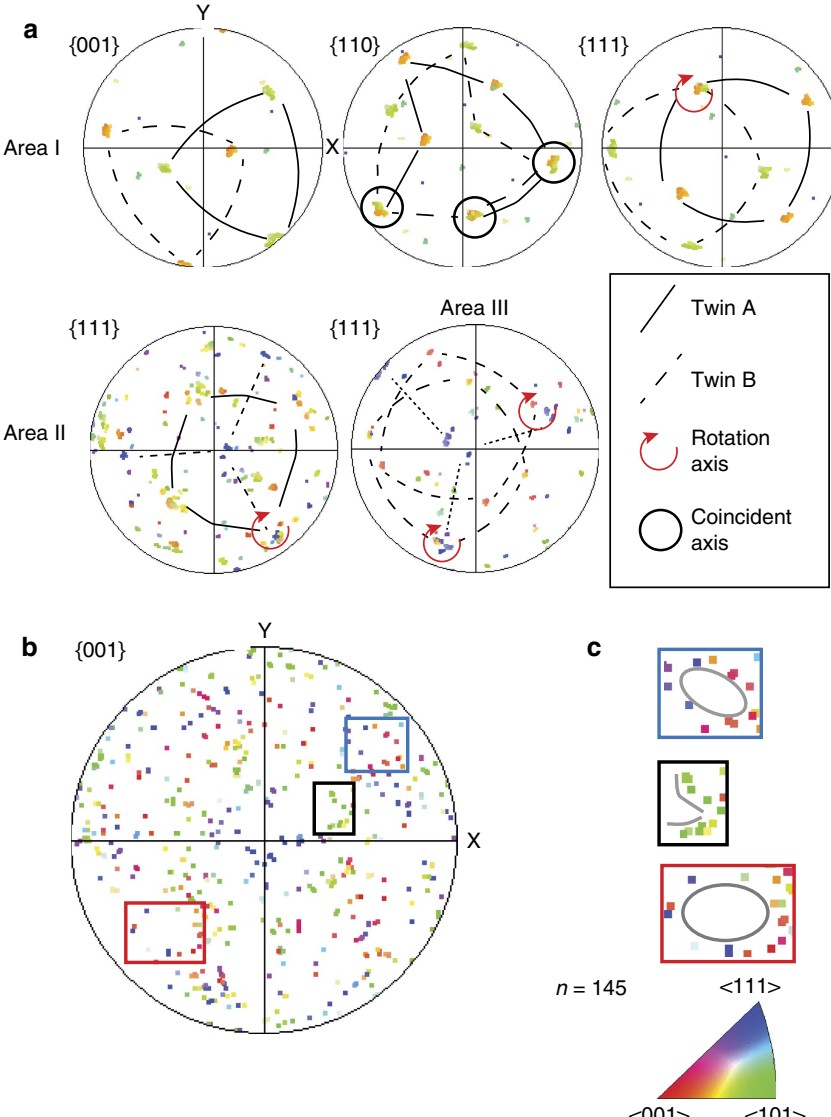

**Figure 6 | Crystallographic analysis of the magnetite corona. (a)** Pole figures for Areas I, II, III from Fig. 5. Each domain is dominated by two orientations with a twinning relationship rotated by 60° around <111>. Up to 3 <110> axes coincide between the twins. **(b)** Magnetite data set for all three areas of the corona (167 data points, one point per grain) showing distinctive non-random distribution with empty areas (some are marked by squares). **(c)** Enlargement of the squared areas shows alignment of orientations in the form of 'rings' (blue and red square) and 'fences' (black square).

pixel (4 × 4 binning), enabling acquisition speeds of 100 patterns per second. During data acquisition all patterns were stored and later reanalysed with optimal conditions for the band detection. The orientation and intensity of 11 Kikuchi bands were compared with those of up to 49 theoretical bands to calculate the orientation of each analysis point. After reanalysis of unindexed points 85–95% of the area could be indexed. Non-indexed areas are dominantly at grain and phase boundaries, especially in the fine-grained magnetite-rich areas. Two data sets were collected on the foil: one on the original foil, and a second one after re-thinning of the magnetite corona by ∼30 nm using FIB. The major part of sulfide inclusion 1 was lost during the re-thinning process. The second data set yielded a better-quality data set at higher resolution for the corona with a much smaller percentage of unindexed areas. Only this data set is used for the interpretation of the magnetite corona. The authors declare that all other relevant data supporting the findings of this study are available on request.

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

## Acknowledgements

S.P. and D.E.J. designed the experiment, P.T. and S.P. carried out the analyses, A.S. carried out the sample preparation, P.T., S.P. and D.J. analysed the data. All authors contributed to writing the manuscript. D.E.J. and S.P. are supported by ARC Future Fellowships. The authors acknowledge the facilities, and the scientific and technical assistance of the Australian Microscopy & Microanalysis Research Facility at the ACMM, The University of Sydney. This is contribution 716 from the ARC Centre of Excellence for Core to Crust Fluid Systems (http://www.ccfs.mq.edu.au) and 1061 in the GEMOC Key Centre (http://www.gemoc.mq.edu.au). D.E.J. is a member of the Deep Carbon Observatory (Reservoirs and Fluxes; https://deepcarbon.net).

## Additional information

**Competing financial interests:** The authors declare no competing financial interests.

