## [Peer Review File · Nature Communications]

Reviewers' comments:

Reviewer #1 (Remarks to the Author):

The manuscript reports the first evidence for diamond formation by redox reactions involving FeNi sulfides. In detail, the authors, using a new experimental approach for natural samples, identified a “quenched” redox reaction from pyrrhotite to magnetite within a diamond of possible eclogitic origin from Orapa mine (Botswana). The work would show definitively a clear link between diamond, sulfides and Fe-oxides. I liked very much the manuscript and I appreciated really much the experimental approach and the details of all steps. I recommend to publish the work as is and I just reported some comments here below.

Line 75. The authors state that the inclusions are of pyrrhotite based on lattice parameters. Sorry, but I could not find any Tables where they reported such lattice parameters. Crystallography of pyrrhotite is so complex and it would be useful for future studies to see more detail about this; it would be sufficient add such data even in the supplementary materials.

Line 182-184. The authors here report a sentence which is not correct: they say that in (26) reference it seems that some olivine are epitaxial with diamonds and some are not. In reference (26), instead, since I am the first author of that work, I can say that we found only random orientation between diamond and olivine. But the reason why we are sure that olivine in that specific case is protogenetic is that we found inside the same diamond different olivine inclusions with the same orientation and we explained this observation with the possibility that such iso-oriented olivines are dissolved fragments from one single preexisting olivine crystal. I say again that this iso-orientation is random with respect to the diamond host and to all other diamonds and olivines studied in that work (i.e. 21 diamonds and 43 inclusions).

Line 188. In principle, I agree with the author statement that the absence of epitaxy does not discount for syngeneses. In the very recent work Bruno et al. (2016) we have demonstrated that, for example, diamond and olivine DO NOT WANT TO STAY in any contact from an energetic point of view (we have calculated the interface energies) and thus any encapsulation must be accidental (Earth and Planetary Science Letters, volume 435, pages 31-35). What I want to say is that the lack of recurrent orientation could mean that the interface energy is identical for any crystal face but this still this does not give you an evidence of protogenesis....but for olivine, combining (a) the random orientation, (b) the extremely low adhesion energy between the diamond and olivine crystal faces, and (c) the evidence of some iso-oriented spatially closed olivines in one single diamond we could say that olivine from Udachnaya kimberlite is protogenetic.

Line 274. Please correct the surnames at the reference F. Nestola et al. The correct names are: “P. Nimis, R.J. Angel, S. Milani, M. Bruno.

Prof. Fabrizio Nestola
Department of Geosciences
University of Padova Italy

Reviewer #2 (Remarks to the Author):

This is a nicely presented manuscript that presents a remarkable level of detail about the processes involved during the formation of a diamond.

Using novel diffraction techniques, the authors demonstrate convincingly the epitaxial growth of diamond on a rim of magnetite coating iron sulfide (pyrrhotite).

The interpretation of this textural and structural information is that the magnetite represents an arrested redox reaction. A further inference is that this redox reaction is also responsible for the formation of the diamond.

Overall, this is a strong manuscript that should be published with some modifications that would strengthen the arguments presented. The issues to be addressed might include the following:

(1) Lines 44-46: Rather than "The extreme overabundance..." of sulfide inclusions in diamonds should be presented more quantitatively if such data exist.

(2) Lines 205-206: "The overabundance of sulfide and magnetite inclusions in diamonds...". Is this statement really correct?

Again, as in (1) above, what quantitative evidence is there of the abundance of magnetite as inclusions in diamond?

(3) I am not convinced of the further inference that the redox reaction from pyrrhotite to magnetite is also responsible for the actual formation of the diamond.

Consider the reaction that might be involved:

e.g. $6\text{FeS} + \text{CO}_2 + 6\text{H}_2\text{O} = \text{C} + 2\text{Fe}_3\text{O}_4 + 6\text{H}_2\text{S}$

(I have simplified the pyrrhotite to be stoichiometric FeS)

Here 6 moles of FeS are needed to produce 1 mole of diamond, and 2 moles of Fe₃O₄.

These relative amounts don't look like the amounts in the arrested reaction that has been identified in the manuscript.

There should be twice as much magnetite as diamond.

(4) I am convinced of the arrested reaction, what I am saying is that it may be an indicator of part of the process of diamond formation, not an actual cause of the bulk of the diamond formation. In other words, during diamond formation there may be many reactions going on, but they are not all reactions that were the main cause of diamond formation.

Minor points are:

(1) Line 36: Magnetite on its own does not define a phase transition. Structures inherited from h-Fe₃O₄ do.

(2) Line 120: should read "Quantification..."

(3) Line 277: Should read "Pyrrhotite..."

Reviewer #3 (Remarks to the Author):

Review of the manuscript NCOMMS-16-01576 by Dr Jacob and co-workers entitled "Redox-freezing and nucleation of diamond via magnetite formation in the Earth's mantle"

The manuscript NCOMMS-16-01576 encompasses results of a detailed and innovative study on polycrystalline diamond aggregate containing eclogite suite inclusions from Orapa diamond mine in Botswana.

General comment is that the scientific quality of this study is very good. Abstract is clear and informative. Experimental approach is original and results are of top quality. The conclusion is intriguing, with the constrain to diamond formation depth. However, lines 138-173, due to their importance, need to be reformulated in a more clear and convincing manner.

Moreover, more comprehensive reference to literature may help drawing conclusions. As for example, in lines 191-207 the authors discuss the controversial presence of sulfides as inclusions in diamonds. An experimental confirmation of the role of sulfides in diamond formation processes is offered by A. V. Shushkanova and Yu. A. Litvin (Doklady Earth Sciences, 2006, Vol. 409A, No. 6, pp. 916-920), and by Yu.N. Palyanov, Yu.M. Borzdov, Yu.V. Bataleva, A.G. Sokol, G.A. Palyanova, I.N. Kupriyanov (Earth and Planetary Science Letters, 2007, 260, 242-256), while a thermodynamic overview of carbon speciation in the asthenosphere is offered by V. Stagno and D.J. Frost (Earth Planet. Sci. Lett., 2010, 300, 72-84).

Further (minor) corrections and suggestions to improve the clarity of the text are described below.

Figure 2 could be displayed together with Figure 1.

Caption to Figure 2 needs to be more explicative (e.g., caption to Figures 2B and 2C is not very clear).

Line 186: *sensu stricto*.

Line 274: 26. F. Nestola, P. Nimis, R.J. Angel, S. Milani, M. Bruno,

Line 315 (and 74 of Supplementary Materials): is the reference number 32?

Line 7 of Supplementary Materials: is the reference number 33?

Jacob et al., point by point answer to reviewers' comments:

Reviewer #1 (Remarks to the Author):

Review of the manuscript “Redox-freezing and nucleation of diamond via magnetite formation in the Earth’s mantle” submitted to Nature Communications.

The manuscript reports the first evidence for diamond formation by redox reactions involving FeNi sulfides. In detail, the authors, using a new experimental approach for natural samples, identified a “quenched” redox reaction from pyrrhotite to magnetite within a diamond of possible eclogitic origin from Orapa mine (Botswana). The work would show definitively a clear link between diamond, sulfides and Fe-oxides. I liked very much the manuscript and I appreciated really much the experimental approach and the details of all steps. I recommend to publish the work as is and I just reported some comments here below.

Line 75. The authors state that the inclusions are of pyrrhotite based on lattice parameters. Sorry, but I could not find any Tables where they reported such lattice parameters. Crystallography of pyrrhotite is so complex and it would be useful for future studies to see more detail about this; it would be sufficient add such data even in the supplementary materials.

We originally identified the pyrrhotite by a combination of (some) d-spacings and element analysis. The element analyses are included in Jacob et al. (2011), Earth and Planetary Science Letters. For the present manuscript, we have relied on the automatic indexation carried out by TKD. This method matches the lattice reflections with a database, thus does not yield individual of d-spacings as one would obtain by TEM. TKD is no match in accuracy with TEM for obtaining d-spacings, but during analysis the best match was obtained with the hexagonal high temperature Fe₉S₁₀ polymorph with a=3.43 and c=5.79 Å. We have rephrased this paragraph to reflect this in a better way.

Line 182-184. The authors here report a sentence, which is not correct: they say that in (26) reference it seems that some olivine are epitaxial with diamonds and some are not. In reference (26), instead, since I am the first author of that work, I can say that we found only random orientation between diamond and olivine. But the reason why we are sure that olivine in that specific case is protogenetic is that we found inside the same diamond different olivine inclusions with the same orientation and we explained this observation with the possibility that such iso-oriented olivines are dissolved fragments from one single preexisting olivine crystal. I say again that this iso-orientation is random with respect to the diamond host and to all other diamonds and olivines studied in that work (i.e. 21 diamonds and 43 inclusions).

Thank you for the clarification, we had indeed misunderstood the authors. Text changed accordingly.

Line 188. In principle, I agree with the author statement that the absence of epitaxy does not discount for syngensis. In the very recent work Bruno et al. (2016) we

have demonstrated that, for example, diamond and olivine DO NOT WANT TO STAY in any contact from an energetic point of view (we have calculated the interface energies) and thus any encapsulation must be accidental (Earth and Planetary Science Letters, volume 435, pages 31-35). What I want to say is that the lack of recurrent orientation could mean that the interface energy is identical for any crystal face but this still this does not give you an evidence of protogenesis....but for olivine, combining (a) the random orientation, (b) the extremely low adhesion energy between the diamond and olivine crystal faces, and (c) the evidence of some iso-oriented spatially closed olivines in one single diamond we could say that olivine from Udachnaya kimberlite is protogenetic.

We fully agree with this interpretation.

Line 274. Please correct the surnames at the reference F. Nestola et al. The correct names are: "P. Nimis, R.J. Angel, S. Milani, M. Bruno.

Corrected

Prof. Fabrizio Nestola
Department of Geosciences
University of Padova
Italy

Reviewer #2 (Remarks to the Author):

This is a nicely presented manuscript that presents a remarkable level of detail about the processes involved during the formation of a diamond.

Using novel diffraction techniques, the authors demonstrate convincingly the epitaxial growth of diamond on a rim of magnetite coating iron sulfide (pyrrhotite). The interpretation of this textural and structural information is that the magnetite represents an arrested redox reaction. A further inference is that this redox reaction is also responsible for the formation of the diamond.

Overall, this is a strong manuscript that should be published with some modifications that would strengthen the arguments presented. The issues to be addressed might include the following:

(1) Lines 44-46: Rather than "The extreme overabundance..." of sulfide inclusions in diamonds should be presented more quantitatively if such data exist.

We agree with the referee that it would be desirable to present such data, however, while this dataset may exist in unpublished company-owned databases, it is unfortunately not available to us.

(2) Lines 205-206: "The overabundance of sulfide and magnetite inclusions in diamonds...". Is this statement really correct?

Again, as in (1) above, what quantitative evidence is there of the abundance of magnetite as inclusions in diamond?

See above, regrettable, we cannot provide a more quantitative dataset.

(3) I am not convinced of the further inference that the redox reaction from pyrrhotite to magnetite is also responsible for the actual formation of the diamond.

Consider the reaction that might be involved:

e.g. $6\text{FeS} + \text{CO}_2 + 6\text{H}_2\text{O} = \text{C} + 2\text{Fe}_3\text{O}_4 + 6\text{H}_2\text{S}$

(I have simplified the pyrrhotite to be stoichiometric FeS)

Here 6 moles of FeS are needed to produce 1 mole of diamond, and 2 moles of Fe₃O₄.

These relative amounts don't look like the amounts in the arrested reaction that has been identified in the manuscript.

There should be twice as much magnetite as diamond.

Answer: The referee is correct in stating that a balanced reaction yields more magnetite than diamond. However, it needs to be kept in mind that the sample presented here may not represent the reaction system quantitatively and completely, nor do we claim that it does. A computed tomography study of this polycrystalline diamond aggregate published earlier (Jacob et al., 2011) shows abundant magnetite associated with the entire sample, albeit still not outweighing the amount of diamond by 2:1. It should also be noted that it is entirely possible that not all magnetite formed in this reaction may not have been sampled. Lastly, while we are not explicitly excluding any other potential diamond forming reactions in our study, we do not have evidence for these.

(4) I am convinced of the arrested reaction, what I am saying is that it may be an indicator of part of the process of diamond formation, not an actual cause of the bulk of the diamond formation. In other words, during diamond formation there may be many reactions going on, but they are not all reactions that were the main cause of diamond formation.

Answer: We agree with the referee, but as stated above, we only have evidence for the reaction that formed magnetite and is described here.

Minor points are:

(1) Line 36: Magnetite on its own does not define a phase transition. Structures inherited from h-Fe₃O₄ do.

corrected

(2) Line 120: should read "Quantification..."

corrected

(3) Line 277: Should read "Pyrrhotite..."

We agree, however this is indeed the original title of the article. Not corrected.

Reviewer #3 (Remarks to the Author):

Review of the manuscript NCOMMS-16-01576 by Dr Jacob and co-workers entitled "Redox-freezing and nucleation of diamond via magnetite formation in the Earth's mantle"

The manuscript NCOMMS-16-01576 encompasses results of a detailed and innovative study on polycrystalline diamond aggregate containing eclogite suite inclusions from Orapa diamond mine in Botswana.

General comment is that the scientific quality of this study is very good. Abstract is clear and informative. Experimental approach is original and results are of top quality. The conclusion is intriguing, with the constrain to diamond formation depth. However, lines 138-173, due to their importance, need to be reformulated in a more clear and convincing manner.

We have rephrased this paragraph and hope that it is now clearer and stronger.

Moreover, more comprehensive reference to literature may help drawing conclusions. As for example, in lines 191-207 the authors discuss the controversial presence of sulfides as inclusions in diamonds. An experimental confirmation of the role of sulfides in diamond formation processes is offered by A. V. Shushkanova and Yu. A. Litvin (Doklady Earth Sciences, 2006, Vol. 409A, No. 6, pp. 916-920), and by Yu.N. Palyanov, Yu.M. Borzdov, Yu.V. Bataleva, A.G. Sokol, G.A. Palyanova, I.N. Kupriyanov (Earth and Planetary Science Letters, 2007, 260, 242-256), while a thermodynamic overview of carbon speciation in the asthenosphere is offered by V. Stagno and D.J. Frost (Earth Planet. Sci. Lett., 2010, 300, 72-84).

Further (minor) corrections and suggestions to improve the clarity of the text are described below.

We have included some of the suggested references.

Figure 2 could be displayed together with Figure 1.

Thank you for this comment, but we prefer to keep these two figures separate

Caption to Figure 2 needs to be more explicative (e.g., caption to Figures 2B and 2C is not very clear).

We have changed the text to provide a clearer caption.

Line 186: sensu stricto.

Corrected

Line 274: 26. F. Nestola, P. Nimis, R.J. Angel, S. Milani, M. Bruno,

Corrected

Line 315 (and 74 of Supplementary Materials): is the reference number 32?

Corrected

Line 7 of Supplementary Materials: is the reference number 33?

Corrected

REVIEWERS' COMMENTS:

Reviewer #1 (Remarks to the Author):

This is the revision of the revised version of the manuscript. I have carefully read the responses of the authors to the referees and I can easily say that they have replied very satisfying to all questions. So I can definitively recommend to publish this work with no further corrections.

Fabrizio Nestola

Reviewer #2 (Remarks to the Author):

This review is a brief response by Rev. 2, requested by Nature Comm. to respond to the revisions made by the authors:

Overall, the authors have taken account of the Reviewers comments pretty well.

But there are two things that I think they have not dealt with:

(1) With regard to the statement about "The extreme overabundance of..." sulfide inclusions, I understand that the authors can only refer to the papers that they have cited (line 45).

However, the sentence late in the paper (line 179) stating

"The overabundance of sulfide and magnetite..." is not justified.

What evidence in the literature is there that there is an overabundance of magnetite???

Sulfides can be said to be overabundant because S is low in the mantle, but sulfide inclusions in diamond are apparently common.

However, Fe is not low in abundance in the mantle! And there is no evidence cited in the text that magnetite inclusions are very common in diamond.

This is important because the authors are claiming in their ms that the pyrrhotite to magnetite reaction is a widespread mechanism. If that's true, magnetite should be very common as an inclusion in diamond. If the author's can't cite any papers in support of this claim, they need to rephrase the sentence on p. 179.

(2) It is ridiculous not to spell pyrrhotite correctly in the list of references. What possible aim can be served by perpetuating a typo in the literature?

Answer to reviewer's comments round 2:

Reviewer #1 (Remarks to the Author):

This is the revision of the revised version of the manuscript. I have carefully read the responses of the authors to the referees and I can easily say that they have replied very satisfying to all questions. So I can definitely recommend to publish this work with no further corrections.

Fabrizio Nestola

Thank you very much

Reviewer #2 (Remarks to the Author):

This review is a brief response by Rev. 2, requested by Nature Comm. to respond to the revisions made by the authors:

Overall, the authors have taken account of the Reviewers comments pretty well. But there are two things that I think they have not dealt with:

(1) With regard to the statement about "The extreme overabundance of..." sulfide inclusions, I understand that the authors can only refer to the papers that they have cited (line 45).

However, the sentence late in the paper (line 179) stating "The overabundance of sulfide and magnetite..." is not justified. What evidence in the literature is there that there is an overabundance of magnetite???

Sulfides can be said to be overabundant because S is low in the mantle, but sulfide inclusions in diamond are apparently common.

However, Fe is not low in abundance in the mantle! And there is no evidence cited in the text that magnetite inclusions are very common in diamond.

This is important because the authors are claiming in their ms that the pyrrhotite to magnetite reaction is a widespread mechanism. If that's true, magnetite should be very common as an inclusion in diamond. If the author's can't cite any papers in support of this claim, they need to rephrase the sentence on p. 179.

We take the referee's comments onboard, there is indeed no support for this claim. The sentence is now removed.

(2) It is ridiculous not to spell pyrrhotite correctly in the list of references. What possible aim can be served by perpetuating a typo in the literature?

We would like to point out that this is not a typo in the title of the cited article by Marx, *Min. Mag.* **38**, 636-638 (1972), but 'pyrrhotine' is the correct English synonym for the mineral and has only been replaced in recent times by the more common use of 'pyrrhotite'. Furthermore, we comply with the Nature Communication policy as given in the guide to authors by citing the title exactly as it appears in the literature.